# Analysis of Engineered Nanoparticles in Seawater Using ICP-MS-Based Technology: From Negative to Positive Samples

**DOI:** 10.3390/molecules28030994

**Published:** 2023-01-19

**Authors:** Olga V. Kuznetsova, Bernhard K. Keppler, Andrei R. Timerbaev

**Affiliations:** 1Vernadsky Institute of Geochemistry and Analytical Chemistry, 119991 Moscow, Russia; 2Institute of Inorganic Chemistry, University of Vienna, 1090 Vienna, Austria; 3Department of Analytical Chemistry and Chemical Ecology, Saratov State University, 410012 Saratov, Russia

**Keywords:** engineered nanoparticles, seawater, ICP-MS, sample preparation, contamination

## Abstract

A growing global emission of engineered nanoparticles (ENPs) into the aquatic environment has become an emerging safety concern that requires methods capable of identifying the occurrence and possibly determining the amounts of ENPs. In this study, we employed sector-field inductively coupled mass spectrometry to assess the presence of ENPs in coastal seawater samples collected from the Black Sea in regions suffering different anthropogenic impacts. Ultrafiltration through commercial 3 kDa membrane filters was shown to be feasible to separate the ENPs from the bulk seawater, and the subsequent ultrasound-mediated acidic dissolution makes the metals constituting the ENPs amenable to analysis. This procedure allowed the ENPs bearing Cu, Zn, V, Mo, and Sn to be for the first time quantitated in seashore surface water, their concentration ranging from 0.1 to 1.0 μg L^−1^ (as metal) and related to the presence of industry and/or urban stress. While these levels are decreased by natural dilution and possible sedimentation, the monitored ENPs remain measurable at a distance of 2 km from the coast. This can be attributed not only to local emission sources but also to some natural backgrounds.

## 1. Introduction

Increasing pollution of aquatic environments is a growing concern with global relevance. Of a large number of chemicals posing considerable risks to aquatic biota and human health, nanomaterials are assumed to be an emerging class of contaminants [1,2,3]. Engineered nanoparticles (ENPs) are produced in large quantities and in a wide range of products, with increasing consumption in a variety of industries (Table 1). Recent estimates [4] predict that the global nanotechnology market will exceed USD 125 billion by 2024. The disposal of ENPs leads to their massive emission into the aquatic environment, where their exact behavior, fate, and ecological implications are still uncertain. It goes without saying that to effectively manage the ENP contamination, an accurate quantification in various marine compartments is imperative. However, these data remain largely elusive due to the analytical difficulties of measuring the ENPs, particularly in a complex seawater matrix. Major barriers exist in two aspects. First, these are low (or very low) concentrations, at which the ENPs arrive and are consumed by the sea. The second obstacle is the extremely high salinity of this type of aquatic sample.

Our recent overview of the relevant literature [5] revealed that nano-sea analysis is still in its infancy and that high-efficiency analytical methods are required to reliably measure the concentrations of ENP pollutants. Another general conclusion is that inductively coupled mass spectrometry (ICP-MS) is the only method capable of targeting seawater concentrations of ENPs. However, the majority of published accounts deal so far with negative (spiked) samples. The only exclusion is the work by Sanchís et al. [6] and de Vega et al. [7], who used a special measurement mode, single-particle (SP) ICP-MS, to differentiate the signal of NPs from the corresponding ion signal. Such an approach makes the direct quantification of ENPs possible, provided that a sample or aerosol is sufficiently diluted. Nonetheless, only a limited assortment of ENPs, such as CeO_2_- and TiO_2_-NPs [6] or TiO_2_- and Pb-based NPs [7], were determined in some coastal seawater samples collected in the metropolitan areas strongly affected by river fluxes. However, consistent measurements of certain target isotopes (e.g., ^48^Ti) and likely of other metals constituting the ENPs (like ^64^Zn) with a quadrupole mass analyzer are hampered by spectral interferences. Apparently, the introduction of high-resolution ICP-MS technology [8,9] may be a step to eliminate this hindrance and streamline the sensitivity of ENP analysis. For instance, by using dynamic-reaction cell methodology, interference-free ICP-MS determination of TiO_2_-, CuO-, Cu-, and ZnO-NPs was proved feasible [10]. However, no attempt to analyze these ENPs in seawater was made in that contribution.

An alternative and highly promising approach recently examined by the authors of [11] is applying sector-field (SF) ICP-MS, following the ultrafiltration isolation of the nanoparticle fraction from matrix salts and background dissolved metal ions. To meet this practice, we have developed a unified protocol for the preparation of seawater samples for examination in a laboratory setting. The absence of ENP loss or transformation, as well as the compromise of measured results, were confirmed using negative, open-sea samples spiked with the relevant ENP concentrations. The objective of the present study was to verify the applicability of an ICP-SFMS-based methodology for the analysis of real seawater samples. These were sourced from different locations of the Black Sea around and at a distance from various industrial facilities and populated urban areas, i.e., with varying anthropogenic pressure.

## 2. Results and Discussion

### 2.1. Sample Treatment Conditions

A schematic illustration of the sample preparation workflow is given in Figure 1 and detailed in the Materials and Methods section below. It is important to note that we used ultrafiltration to separate the ENPs from bulk seawater samples so that the retained particles are then brought into solution after dissolution directly in the filtration unit and analyzed as the respective metal ions [11]. This approach allows for easy discrimination of signals from nanoparticulate matter and an ionic background. Previously, it was shown that commercial membrane filters prove useful for nanoparticle separation [11,12,13], and the 3 kDa filter units seem to be preferable than other molecular mass cut-offs [11,14,15] (at least for the particles larger than 10 nm in diameter). However, it is essential to precondition the filter membrane, e.g., with 0.1 M copper nitrate [11,13], to avoid ionic metal adsorption. Since such treatment might compromise the determination of copper-based NPs, we prefer a nickel nitrate solution in this case. It should be noted that copper-containing nanomaterials find more common industrial use and are expected to occur in seawater at higher concentrations than Ni-NPs (see also Section 2.3). The fact that nickel nitrate can well substitute Cu(NO_3_)_2_ was verified experimentally, and the comparative recovery data for the test ENPs are shown in Table 2.

### 2.2. Metal Concentrations in Seawater

Next it was important to analyze the seawater samples under examination to define the levels of metals that appear to constitute the ENPs and it this way, be relevant regarding the expected ENP contents. Another purpose of these measurements was to assess the method’s detectability as the basis for a more proper data interpretation for ENPs (see below). Table 3 summarizes the analytical results, together with the limits of detection (LODs) defined as specified in the footnote. Notably, sample dilution to avoid interference from the high-salt matrix has not impeded quantification. This is no analytical challenge when using ICP-SFMS due to the method’s combination of high sensitivity and relative freedom from spectral interferences offered by high resolution capabilities.

As can be expected, in the coastal seawater collected close to the region known for industrial production activities, the concentration levels are markedly higher than for the seaside area with lesser inputs from anthropogenic sources (cf. S0 and K0 data), particularly for common metals such as iron, zinc, or chromium. Concentrations measured in the samples taken at a noticeable, two kilometer distance from the coastline are in most cases lower than the seashore concentrations, though they are still not free from contamination from different industrial or natural sources. This follows from a comparison with the data for distant open-sea water sampled far from natural or anthropogenic influences (e.g., 0.020 and 0.012 µg L for Cd and Pb, respectively [16]).

### 2.3. ENPs in Seawater

The application of ICP-SFMS in combination with ultrafiltration isolation identified the presence of ENPs and even made it possible to quantify some particles. It should be mentioned that only those samples containing ENPs at concentrations higher than three times the LOD were considered positive (field blank data are shown in the Materials and Methods section). More specifically, Ag- and Ti-based NPs were found to occur in the contaminated coastal seawater at the level of limits of quantification, viz., 0.04 and 0.33 μg L^−1^ (as metal), respectively. Of nanotechnology-related materials, these are among the most commonly employed products (see Table 1), and in addition, they may escape from wastewater treatment plants into the aquatic environment [17]. However, previous studies have shown that the Ag-NPs tend to dissolve in seawater [6,15], while for titanium-bearing NPs, one of the main sources in seawater, sunscreen lotions, is barely a factor beyond the beach season. The origin of detectable Sb-NPs (about 0.04 μg L^−1^) is not clear, but potential sources might be from various industrial as well as natural sources (e.g., weathering of the stibnite ore). On the other hand, no nanolead was detected, although it was reported that particulate lead may exist in freshwater at least [18], possibly as a result of the detachment of lead corrosion products. The most plausible explanation is due to a high salt content exerting a considerable dissolving action on Pb nanomaterials [7].

The suspended ENPs of Cu, Zn, V, Mo, and Sn were measured not only within the metropolitan area but also in the area with little anthropogenic pressure (see the left-hand bars in Figure 2), and this is the key outcome of the present study. The fact that no such nanomaterials were detected in previous research based on using SP-ICP-MS [6,7] can be rationalized by taking into account the substantial sample (or aerosol) dilution used to reduce the introduction of salt into the plasma (let alone the order-of-magnitude sensitivity advantage of the SF system over quadrupole-based mass spectrometers). In our approach, samples are analyzed undiluted, and furthermore, the nanoparticulate analytes could be even enriched when large-volume filtration units are employed. For the mass-produced and frequently employed Zn- and Cu-NPs, the concentrations in coastal samples reach 1.0 μg L^−1^ and higher. However, similar levels found in the sea section without any expected industrial impact demonstrate that the presence of NPs produced through natural pathways should not be neglected. It is also worth noting that the method cannot differentiate between Cu(I)- and Cu(II)-bearing nanomaterials. The unusually high concentrations of nano-Mo in the samples taken in the vicinity of urbanized areas can be attributed to its industrial usage, e.g., in steel products [19].

Figure 2 shows the ENP concentrations in the samples collected at increased distances from the coast. Regardless of the ENP type, the dependences are of the same character, and such longitudinal distribution is presumably triggered by natural dilution in combination with agglomeration/aggregation effects. Similarly, for the total metal concentrations, the content of each sort of ENP, while fading with distance, is still high enough to make the determination in offshore samples possible. Therefore, the issue to be addressed in future studies is whether the monitored concentration levels might pose an environmental risk. Special attention from the ecotoxicological viewpoint should also be paid to the size and exact composition of detected particles, neither of which are discernible with ICP-SFMS. Neither can the method distinguish between anthropogenic and natural nanomaterials unless the isotopically labeled ENPs are involved [20].

## 3. Materials and Methods

### 3.1. Sample Location and Collection

All samples were taken in May 2022 from the Black Sea. Sampling locations are schematically represented in Figure 3. The studied areas are with different levels of anthropogenic pressure, a bay area vs. a close proximity of a highly urbanized city with notable industries, a large port facility and cover a distance up to 2000 m from the shore. Under flat sea conditions, the collected surface seawater samples were filtered through 0.45 μm polycarbonate membrane filters (Sigma-Aldrich, St. Louis, MO, USA) to remove larger particulate matter, slightly acidified with ultrapure nitric acid (65%, Merck, Germany; to pH 7.5), and placed in acid-cleaned low-density polyethylene bottles (Nalgene General Oceanics, Miami, FL, USA). For preservation, the samples were refrigerated in the dark at 4 °C. Such sample treatment/storage conditions were proven to ensure particle stability prior to analysis [11]. Used as a field blank, Milli-Q water was stored, treated, and analyzed in the same way as the real samples [6].

### 3.2. Chemicals and Materials

Reference ENPs, silver, titanium dioxide, and zinc oxide NPs (of <100, 21, and <50 nm nominal sizes, respectively) were purchased from Sigma-Aldrich. Each type of particle was suspended in 0.5 M NaCl by vortexing for 5 min to provide standard suspensions of 0.81, 120.5, and 27.7 mg L^−1^ Ag, TiO_2_, and ZnO, respectively. Primary metal standards (10 mg L^−1^) used to prepare mixed standard solutions and the internal standard (High-Purity Standards, Charleston, SC, USA) were diluted appropriately with 3% HNO_3_, prepared from nitric acid and ultrapure water. Amicon Ultracel 0.5 mL units with cut-off molecular masses of 3 kDa used for ultrafiltration experiments were obtained from Millipore (Molsheim, France). The units were preconditioned with 0.1 M copper or nickel nitrate (Sigma Aldrich) and rinsed with Milli-Q water upon centrifugation before use.

### 3.3. Sample Preparation

Individual seawater samples were first taken for total metal analysis carried out after a 10-fold dilution with 3% HNO_3_ (as validated in [16]). Then the undiluted samples were vortexed for 5 min, and aliquots were filtrated through the filters with a 3 kDa cut-off for 15 min at 10,000 rcf (Sigma 1–14k, Sigma Laborzentrifugen GmbH, Osterode am Harz, Germany) to separate the nanosized fraction of the particulate matter. The latter was washed twice with water by repeating the centrifugation to remove dissolved metal species, and the remaining particles were treated with 30% HNO_3_ (upon a 10 min sonication). The resulting solution was subjected to short ultrafiltration, and the filtration membrane was washed with water upon centrifugation. Finally, the combined filtrates were diluted to achieve an acid concentration of 3–5%. Each sample was analyzed at least three times. A field blank was carried out to check for potential sample contamination, and the results are shown in Table 4. Positive blank concentrations were subtracted from real samples.

### 3.4. ICP-SFMS Analysis

The metal concentrations in acidified samples were determined using an Element 2 instrument (Thermo Fisher Scientific, Waltham, MA, USA), operating in low (*R* = 300) or medium resolution setting (*R* = 4000), and the following instrumental settings: plasma gas flow, 14 L min^−1^; auxiliary gas flow, 0.9 L min^−1^; nebulizer gas flow, 0.9 L min^−1^; analyzed sample flow, 0.8 L min^−1^; RF power, 1250 W; and dwell time, 20 ms. The internal standard ^115^In was analyzed to correct for non-spectral interferences during analysis. As was shown in our previous research [16], under the mentioned conditions, the isotopes of the ENP-constituting metals can be monitored in seawater without spectral interferences. This is particularly important in the case of titanium, whose signal cannot be fully released from the interference of Ca when using quadrupole-based ICP-MS [7].

## 4. Conclusions

Whatever measurement mode or mass analyzer is used, ICP-MS is the one and only method suitable for the quantification of very low seawater concentrations of ENPs. We demonstrated the potential and applicability limits of ICP-SFMS for testing coastal seawater for these as-yet unidentified targets. A number of ENPs, such as those bearing Cu, Zn, V, Mo, and Sn, were reliably determined by employing a versatile analytical protocol for the storage, preparation, and measurement of seawater samples. The benefits of the developed technique are due to the high sensitivity of the SF-based system and the ability to analyze the nanoparticulate matter without sample dilution. Some assumptions are made about the origin of target nanomaterials in tested aquatic areas, accounting for anthropogenic and natural contributions. Our upcoming research will be aimed at comparing the performance of ICP-SFMS as a nano-sea analysis tool with that of triple-quadrupole and SP-ICP-MS techniques.

## Figures and Tables

**Figure 1 molecules-28-00994-f001:**
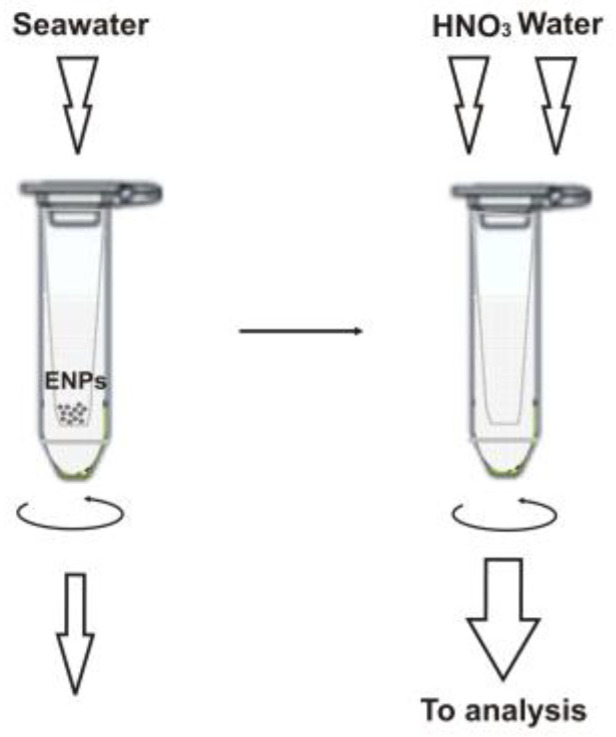
Sample preparation for the ICP-SFMS analysis.

**Figure 2 molecules-28-00994-f002:**
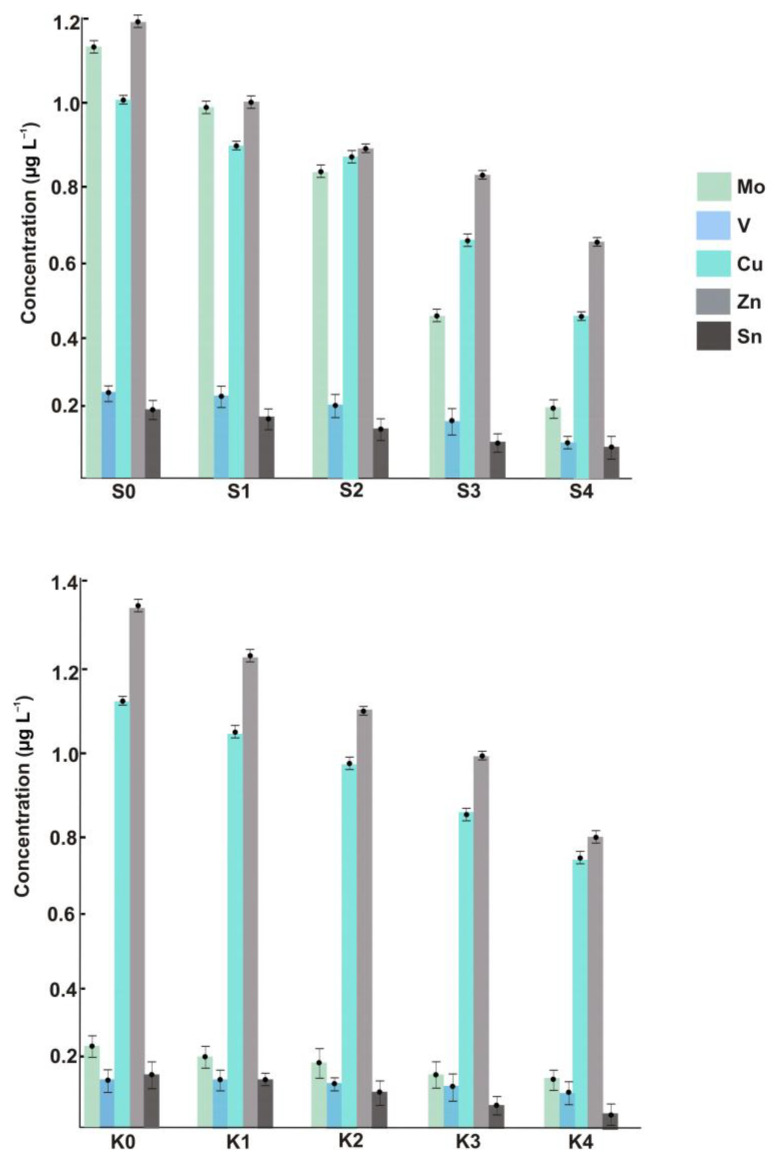
Concentrations of ENPs across different sample locations of the Black Sea.

**Figure 3 molecules-28-00994-f003:**
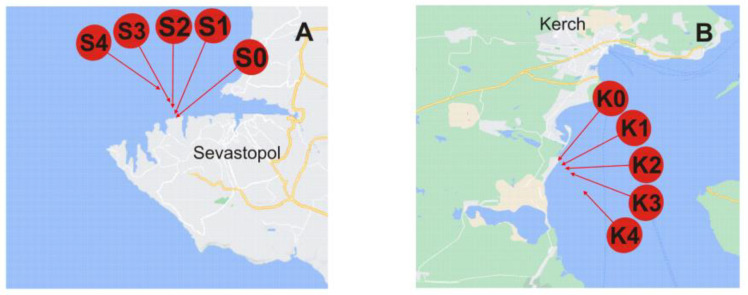
Location of sampling sites across the Black Sea. Sampling stations legend: (**A**) S0, S1, S2, S3, and S4 from the Sevastopol City area are taken at the same distances from the coast. (**B**) K0, K1, K2, K3, and K4 from Kerch Bay at a distance of ~5, 100, 200, 500, and 2000 m from the coast, respectively.

**Table 1 molecules-28-00994-t001:** Major types and modern uses of ENPs ^a^.

ENPs	Main Fields of Application
Ag	Medicine, textiles, cosmetics, home appliances, environment
TiO_2_	Cosmetics, construction, textiles, renewable energies
SiO_2_	Construction, textiles, automotive, cosmetics
ZnO	Cosmetics, skin care
CuO	Batteries, sensors, superconductors, coatings
Cu_2_O	Coatings and paints
CeO_2_	Fuel additives, abrasive material
Quantum dots	Electronics
Au	Electronics, medicine
MoO_3_	Batteries, fuel additives, coatings, catalysts
SnO_2_	Batteries, coatings, catalysts, sensors

^a^ See https://www.futuremarketsinc.com/the-global-market-for-nanomaterials-2010-2022 (Accessed on 2 December 2022) for more detail.

**Table 2 molecules-28-00994-t002:** Recovery of selected ENPs from seawater after using the different preconditioning agents.

ENPs ^a^	Recovery (%)	
	0.1 M Copper Nitrate	0.1 M Nickel Nitrate
Ag-NPs	85 ± 8	88 ± 9
Ti-NPs	107 ± 12	109 ± 11
Zn-NPs	109 ± 13	98 ± 11

^a^ 0.5, 4.6, and 41.2 µg L^−1^ Ag, Ti, and Zn (or 10 times the limit of quantification), respectively.

**Table 3 molecules-28-00994-t003:** Concentrations and limits of detection of metals in seawater (µg L^−1^ ± σ; *n* = 3) ^a^.

Metal	Measured Value ^b^				LOD ^c^
	S0	S4	K0	K4	
Ag	0.53 ± 0.03	n.d. ^d^	n.d.	n.d.	0.009
Bi	0.38 ± 0.02	0.36 ± 0.02	0.42 ± 0.03	0.46 ± 0.04	0.0013
Cd	0.21 ± 0.01	0.21 ± 0.01	0.21 ± 0.01	0.22 ± 0.01	0.004
Co	n.d.	n.d.	n.d.	n.d.	0.014
Cr	2.28 ± 0.08	0.64 ± 0.04	0.64 ± 0.08	0.46 ± 0.08	0.029
Cu	2.07 ± 0.05	0.80 ± 0.07	2.83 ± 0.12	0.95 ± 0.06	0.06
Fe	32.7 ± 0.5	5.94 ± 0.08	7.99 ± 0.21	1.86 ± 0.06	0.06
Mn	0.13	0.11	0.08	0.11	0.05
Mo	5.60 ± 0.12	0.76 ± 0.02	5.07 ± 0.28	0.87 ± 0.02	0.007
Ni	n.d.	n.d.	n.d.	n.d.	0.13
Pb	0.78 ± 0.06	0.58 ± 0.07	0.73 ± 0.06	0.55 ± 0.04	0.005
Sb	0.37 ± 0.01	0.31 ± 0.05	0.42 ± 0.04	0.39 ± 0.03	0.008
Sn	0.36 ± 0.01	0.21 ± 0.03	0.46 ± 0.03	0.26 ± 0.03	0.009
Ti	0.65 ± 0.04	0.44 ± 0.03	0.74 ± 0.07	0.44 ± 0.07	0.079
V	0.44 ± 0.03	0.11 ± 0.004	0.28 ± 0.02	0.11 ± 0.01	0.009
Zn	24.0 ± 0.4	5.64 ± 0.20	11.7 ± 0.4	4.50 ± 0.30	0.10

^a^ Ten times diluted with 3% HNO_3_. ^b^ For sampling station codes here and below, see Materials and Methods. ^c^ Method detection limit, defined here as three times the standard deviation of blank signal intensity. ^d^ n.d. = not detected.

**Table 4 molecules-28-00994-t004:** Metal concentrations in field blank sample (µg L^−1^ ± σ; *n* = 3) ^a^.

Metal	Measured Value
Cr	0.11 ± 0.05
Cu	0.40 ± 0.05
Fe	0.72 ± 0.05
Mn	0.20 ± 0.05
Zn	0.70 ± 0.10

^a^ Other metals were not detected.

## Data Availability

Not applicable.

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
