# Peer review of "Analysis of Engineered Nanoparticles in Seawater Using ICP-MS-Based Technology: From Negative to Positive Samples"

_molecules, 2023, doi:10.3390/molecules28030994_

Round 1

Reviewer 1 Report

Paper title: Analysis of Engineered Nanoparticles in Seawater Using ICP- 2 MS-Based Technology: from Negative to Positive Samples

In this study, The authors employed sector-field inductively coupled mass spectrometry to assess the presence of ENPs in coastal seawater samples collected from the Black Sea in regions suffering different anthropogenic impact. Ultrafiltration through commercial 3 kDa membrane filters was shown feasible to separate the ENPs from the bulk seawater and the subsequent ultrasound-mediated acidic dissolution makes the metals constituting the ENPs amenable to analysis. This procedure allowed the ENPs bearing Cu, Zn, V, Mo, and Sn to be for the first time quantitated in seashore surface water.

The paper could be interested for readers. Some mnior revisions are needs before acceptance.

1. Table 1 should not placed in introduction part. The authors can provide the contents of the table inside the introduction and no need to mention a table

2. In Introduction part, please highlight the strength of the current paper.

3. Figure 1, the title is not appropriate, it is not a scheme

4. Titles of tables are too short

5. In table 3, I did not see the meanings of the mentioned abbreviations

6. What are the limitations of the conducted investigation?

7. Please improve the conclusion part

Author Response

  1. Table 1 should not placed in introductory part. The authors can provide the contents of the table inside the introduction and no need to mention the table

In our opinion, such an alteration would make the introduction less comprehensible while the information presented in a tabular form gives an interested reader a clear view on the types and uses of engineered nanoparticles. 

  1. In introduction part, please highlight the strength of the current paper

It is highlighted by stating that the ICP-SFMS-based methodology is applicable for the analysis of real seawater samples.

  1. Figure 1, the title is not appropriate, it is not a scheme

Corrected.

  1. Titles of table are too short

Modified.

  1. In Table 3, I did not see the meanings of the mentioned abbreviations

Added to footnotes

  1. What are the limitations of the conducted investigation?

They are specified in the last paragraph of R & D part. Our method cannot discern the size and exact composition of detected particles, as well as anthropogenic and natural nanomaterials.

  1. Please improve the conclusion part

Done.

Reviewer 2 Report

Kuznetsova et al established an ICP-MS method for ENP analysis in seawater. This method can be used for both negative and positive samples. This topic is interesting and fits the scope of this special issue. I recommend this manuscript to be published after major revision.

Specific comments:
L5: “1” not “!”. The same in Supplementary Materials.

L48: the citation style of reference 7 is different from others. “de Vege et al” is suggested.

L60: it is not clear why ENPs were not analyzed in seawater in previous studies. what’s the hinder? The complicated matrices of seawater that varies from freshwater?

L90: Why different concentrations of Ag, Ti and Zn were selected? For Ag, Ti and Zn, different concentrations of individual targets should be tested. The LODs for Ag-, Ti-, and Zn-NPs are missing.

L103: one sample was tested in triplicate? I can not find this information.

It seems that the authors tested the recoveries of Ag-, Ti-, and Zn-NPs. Yet, how they measure the concentrations of Cu-, Zn-, V-, Mo-, and Sn-NPs? And how about their LODs?

Section 3.3: information on the instruments used for sample preparation is missing.

L215-216: the authors can not conclude that “ICP-MS is one and only one method suitable for the quantification of very low seawater concentrations of ENPs” because they did not compare other methods in this present study. 

Author Response

L5: “1” not “!”. The same in Supplementary Materials.

Corrected. Please note that Supplementary Materials were merged with the main text to adjust the paper size to journal standards.

L48: the citation style of reference 7 is different from others. “de Vege et al” is suggested.

Accepted.

L60: it is not clear why ENPs were not analyzed in seawater in previous studies. what’s the hinder? The complicated matrices of seawater that varies from freshwater?

This issue is discussed in the second paragraph of Section 2.3. The disadvantage of procedures used in previous studies is that sample or aerosol was substantially diluted to reduce the introduction of salt into the plasma.

L90: Why different concentrations of Ag, Ti and Zn were selected? For Ag, Ti and Zn, different concentrations of individual targets should be tested. The LODs for Ag-, Ti-, and Zn-NPs are missing.

We adjusted the concentrations of these particles to the respective limits of quantification determined in our previous contribution (ref. 11). This information is added to Table 2.

L103: one sample was tested in triplicate? I can not find this information.

We believed that was clear from “n = 3” in the tables but anyway mentioned this information in the revised text (Section 3.3).

It seems that the authors tested the recoveries of Ag-, Ti-, and Zn-NPs. Yet, how they measure the concentrations of Cu-, Zn-, V-, Mo-, and Sn-NPs? And how about their LODs?

We measured these similarly as the concentrations of metals constituting the nanoparticles (LODs are given in Table 3).

Section 3.3: information on the instruments used for sample preparation is missing.

We added the type of the centrifuge that was the only instrument used for sample preparation.

L215-216: the authors can not conclude that “ICP-MS is one and only one method suitable for the quantification of very low seawater concentrations of ENPs” because they did not compare other methods in this present study. 

This is true but the correctness of the statement in quotation marks follows from our recent review paper (ref. 5).

Round 2

Reviewer 2 Report

Accept.